# Flicker Noise in Resistive Gas Sensors—Measurement Setups and Applications for Enhanced Gas Sensing

**DOI:** 10.3390/s24020405

**Published:** 2024-01-09

**Authors:** Janusz Smulko, Graziella Scandurra, Katarzyna Drozdowska, Andrzej Kwiatkowski, Carmine Ciofi, He Wen

**Affiliations:** 1Faculty of Electronics, Telecommunications and Informatics, Gdańsk University of Technology, Narutowicza 11/12, 80-233 Gdańsk, Poland; katarzyna.drozdowska@pg.edu.pl (K.D.); andrzej.kwiatkowski@pg.edu.pl (A.K.); 2Department of Engineering, University of Messina, 98166 Messina, Italy; graziella.scandurra@unime.it (G.S.);; 3College of Electrical and Information Engineering, Hunan University, Changsha 410082, China; he_wen82@126.com

**Keywords:** flicker noise, gas sensors, noise measurements, classification algorithms

## Abstract

We discuss the implementation challenges of gas sensing systems based on low-frequency noise measurements on chemoresistive sensors. Resistance fluctuations in various gas sensing materials, in a frequency range typically up to a few kHz, can enhance gas sensing by considering its intensity and the slope of power spectral density. The issues of low-frequency noise measurements in resistive gas sensors, specifically in two-dimensional materials exhibiting gas-sensing properties, are considered. We present measurement setups and noise-processing methods for gas detection. The chemoresistive sensors show various DC resistances requiring different flicker noise measurement approaches. Separate noise measurement setups are used for resistances up to a few hundred kΩ and for resistances with much higher values. Noise measurements in highly resistive materials (e.g., MoS_2_, WS_2_, and ZrS_3_) are prone to external interferences but can be modulated using temperature or light irradiation for enhanced sensing. Therefore, such materials are of considerable interest for gas sensing.

## 1. Introduction

Gas sensing is a continuously developing research area utilizing new materials and technologies to enhance gas sensors’ properties [1,2]. A popular group of sensors is chemoresistive gas sensors that change their resistance at various ambient atmospheres [3,4,5]. The main driving forces of enhancing resistive gas sensors are their improved sensitivity, selectivity, and response stability in time (drift in time reduction). One such method utilizes resistance fluctuations at a low-frequency range that depends on ambient atmosphere. The technique was named fluctuation-enhanced sensing (FES) and was proposed over two decades ago [6,7,8]. The FES method proved its advantages in numerous experiments by detecting a few gases with a single gas sensor or detecting an adsorbed single gas molecule. These results were thoroughly reviewed elsewhere [9].

Resistive gas sensors utilize grainy material, and adsorbed gas molecules modify the potential barrier between the grains and, as a result, change their resistance between the terminals. Low-frequency noise (flicker noise, 1/f-like noise) depends on adsorption–desorption events of gas molecules with the gas sensing material. These events modulate the potential barrier between the grains, resulting in low-frequency resistance fluctuations between the sensor terminals. Both physical quantities (resistance and fluctuations) are independent and secure information about the ambient atmosphere.

Resistance fluctuations are observed within the low-frequency range, typically up to tens of kHz, and do not depend on gas sensor bias conditions specific to the applied measurement setup [10]. We estimate the power spectral density of resistance fluctuations to gather more information about the sensors’ ambient atmosphere. Its intensity and changing slope versus frequency depend on ambient atmosphere composition [11,12]. The main component of the observed 1/f noise uses the Hooge semiempirical formula [13]:(1)$$S_{I}\left(f\right)/I^{2}=S_{U}\left(f\right)/U^{2}=S_{R}\left(f\right)/R^{2}=\alpha_{H}/Nf$$
where *S_I_*(*f*), *S_U_*(*f*), and *S_R_*(*f*) are the current, voltage, and resistance spectral densities of the sample polarized by *I* current, *U* voltage, and DC resistance *R*, respectively; *N* is the number of carriers in the sample; and *α_H_* is the Hooge constant representing flicker noise intensity in the studied sample. When we consider 1/*f* noise for gas sensing, resistance power spectral density *S_R_*(*f*) is independent of the sample bias conditions in the applied measurement setup and susceptible to the sensor’s ambient atmosphere. It means that scaling adequate to the measurement setup is necessary, as presented elsewhere [10].

Except for the 1/*f* noise component, the slope of power spectral density *S*(*f*) can change locally because of the adsorption–desorption events of characteristic time constants *τ*_i_. These components generate additional low-frequency noise items called Lorentzians that can be generated separately for different gases of various adsorption–desorption rates and of different intensities *A_i_*:(2)$$S\left(f\right)\propto \;\sum_{i=1}^{N}A_{i}\frac{\tau_{i}}{1+{\left(2\pi f\tau_{i}\right)}^{2}}.$$

Its presence manifests as a plateau in the 1/f-like spectrum (Figure 1) and can be exposed as a local maximum by plotting a product *S*(*f*)·*f*. Lorentzians can detect gases even when transferring a single electron [14]. Time constants *τ*_i_ can be characteristic values for selected gases when low-frequency noise is generated in tiny and repeatable structures like a single layer of graphene [15]. Moreover, Lorentzians preserve their characteristic frequencies *f*_i_ = 1/2π*τ*_i_ even if the graphene layer is of a much more significant area [16].

When the 1/*f*-like noise dominates over white noise within a frequency range up to tens of kHz, we can identify a few Lorentzians at the maximum to enhance gas sensing properties. The FES method can improve gas sensing properties using a single gas sensor if we can measure and identify its existing low-frequency noise components. The applied resistive sensing materials have various physical properties, from metallic to semiconducting. They can be modified using dopants (e.g., noble metal additives, etc.) [17,18,19] or modulated via an operating elevated temperature or irradiation [20,21]. Their DC resistances determine the way of low-frequency noise measurements and further analysis. These issues are further considered in detail in our studies to propose technical solutions for low-frequency noise measurements.

Recent advances in gas sensing, using doped materials, improved gas sensitivity by considering DC resistances only. A change of DC resistance by a few multiples was observed for selected gases (C_3_H_6_O, CO gases [22]) for noble metal doping of the WS_2_ sensor. Only a few percent was observed for the Cu doping of the LaFeO_3_ sensor (CO_2_ gas [23]). The doping shortened the response/recovery to a few hundred seconds; however, the FES method is still very attractive because we observe a change in noise power spectral intensities that can be as large as a few orders of magnitude at very low gas concentrations [9].

The paper is organized as follows: Section 2 presents the materials and methods used for resistive gas sensors, low-noise measurement setups, and data analysis methods; Section 3 presents the exemplary results of noise measurements; Section 4 discusses the main issues of noise measurements; and finally, Section 5 summarizes the results of our investigation.

## 2. Materials and Methods

### 2.1. Resistive Gas Sensors

Resistive gas sensors utilize various materials, typically grainy metal oxides (MOS—metal oxide sensors), e.g., ZnO, SnO_2_, TiO_2_, WO_3_, CuO, etc. of different morphologies and additives to enhance their gas selectivity and sensitivity [1,4,24]. The sensors can comprise pristine or hybrid materials [25], nanoparticles (e.g., Au) functionalized by organic ligands [26], or two-dimensional materials operating as field effect transistors (FETs) [3,27,28,29,30]. The sensors made of metal oxides operate at elevated temperatures, up to a few hundred degrees Celsius (typically 100–450 °C), to accelerate adsorption–desorption rates (Figure 2A). It means that the sensors consume more energy and can be used to determine the gas composition of hot fumes. There is a new commercial generation of such sensors with reduced energy consumption of a factor of about ten times by applying MEMS technology in recent years [31]. Such sensors can be applied in portable devices like smartphones to monitor local atmosphere conditions or even used in the biomedical field [32,33,34].

Another possible configuration utilizes back-gated FETs where the sensing material (e.g., a single layer of graphene) is situated between the metal electrodes, source, and drain, and the substrate works as the third gate electrode (Figure 2B) [15,35,36]. We utilize two-dimensional (2D) materials (e.g., graphene, 2D metal dichalcogenides: MoS_2_, WS_2_) for gas sensing because of their high volume-to-active area ratio [27,28]. Two-dimensional materials have various conducting properties, from metallic to semiconducting. It means we can modulate their gas sensing properties by changing a gate voltage and modifying their energy gap or applying UV irradiation to generate weakly bounded oxygen ions, replaced by detected gas molecules [15,16].

The events of gas adsorption–desorption occur within the structure imperfections of the applied 2D material [37,38]. Therefore, their existence is crucial for gas sensing. Different defects can also be modified using dopants, e.g., by using noble metals exhibiting a drastic increase in gas selectivity [39]. It was experimentally observed that Au nanoparticles attenuate flicker noise in graphene and ZnO heterostructures due to catalytic and localized surface plasmon resonance effects [40,41,42]. Additionally, an intense light-matter interaction means a light-modulated FES method, which makes this method even more attractive.

Graphene was one of the first 2D materials used for gas sensing and is continuously developing [37,43]. This material can be used for gas sensing as a single-layer flake, in back-gated FET configuration [15,16], or as ink-printed layers of graphene flakes [44,45]. Its gas-sensing properties depend on structural imperfections appearing during the graphene growth process. The defects can be modified using nanoparticle decoration (e.g., TiO_2_, ZnO, and SnO_2_) [46,47]. It was reported that graphene-based gas sensors can detect numerous gases (e.g., organic solvents, NO_2_, NO, SO_2_, H_2_S, H_2_O, etc., [48]). Graphene is characterized by its close to metallic electrical properties (high electrical and thermal conductivity). Therefore, its DC resistance fluctuations can be measured using low-noise voltage amplifiers applied for noise sources of DC resistances about tens of kΩ at the maximum.

The low-frequency noise depends on FET bias voltages, and the results observed for various biases can be used for enhanced gas sensing [49]. There are unusual and unexplained gate voltage dependence of noise and mechanism of fluctuations in the atmosphere of different gases and also at different gate voltages [50]. Additionally, we can apply UV irradiation to modulate gas sensing properties by generating oxygen photo-ions weakly bound to graphene and, therefore, more easily replaced with ambient gas molecules. This effect leads to enhanced gas sensitivity [16]. The graphene layer is somewhat prone to UV irradiation, but wavelengths below 220 nm, which are too short, can damage the 2D structure.

Moreover, the graphene situated over the dielectric SiO_2_ layer is prone to adsorbing humidity from ambient air and reducing sensing abilities. This process is relatively fast and requires sensor refreshment by heating in a vacuum to remove adsorbed humidity. It was proved experimentally that such detrimental effect of rapid aging was reduced when the graphene layer was placed on, e.g., GaN cap or conducting n-doped Si [40,51].

Extensive experimental and theoretical studies were run on gas sensors utilizing MoS_2_ semiconducting material (Figure 3), including the FES method [30,52]. This material is promising for gas sensing applications because of its semiconducting properties with a bandgap of 1.23 eV in bulk, which is similar to Si. The MoS_2_ gas sensor can be modified by decorating it with nanoparticles of noble metals to obtain high sensitivity [53] or drastically improving selectivity to some gases [39]. Similar results of enhanced selectivity were observed for MoS_2_ nanosheets decorated with MoO_3−x_ impurities [54]. Gas selectivity is reached via nanoparticle decoration of chosen MoS_2_ imperfections at its edges or on 2D material surface. Depending on their bounding energy, some defects are more prone to be decorated by nanoparticles. This selective process means we can modify the sensor to enhance its selectivity.

Additionally, MoS_2_ is a photocatalytic material that can be modulated using UV irradiation to increase gas sensitivity and reduce response and recovery times (e.g., for NO_2_ gas [55,56]).

### 2.2. Mechanisms and Intensity of Low-Frequency Noise in Selected 2D Materials

Using the FES method, we focused on two selected 2D materials (graphene and MoS_2_) for gas sensing. These two materials are gas-sensitive, and their 1/f noise can be measured in different conditions [57]. Power spectral density (PSD) of current fluctuations *S*_I_(*f*) normalized to the square of the bias current *I*^2^, see Equation (1), can be a few orders greater for the MoS_2_ sensor than for graphene but is more difficult to measure because of very low bias current *I*. The DC resistance of the MoS_2_ gas sensor can reach a GΩ, and it makes noise measurements very difficult and prone to existing external interferences.

The 1/f noise was observed in the presented 2D materials during numerous experiments and was reported elsewhere [56,57,58,59,60,61,62,63]. Flicker noise in graphene and MoS_2_ depends on frequency as *S*(*f*)~1/*f*^α^ with almost a similar coefficient range *α* = 0.75–1.25. The MoS_2_-based FET exhibits weak gate voltage dependence [64]. Low-frequency fluctuations generated within a graphene structure do not follow the McWhorter model. Recently published experimental results suggest that the main component of flicker noise in graphene is related to mobility and scattering mechanisms rather than concentration fluctuations or trapping-de-trapping processes [61]. When exposed to some organic gases, the single-layer graphene FET generated Lorentzians of the frequencies *f*_i_ characteristic for selected gases [15,16]. Similar results were observed for a composition of graphene flakes and TiO_2_ nanoparticles [45].

Flicker noise in graphene scales with a square of bias current *I*^2^ and inversely with its area *A*^−1^. The normalized low-frequency noise amplitude *A*·*S*(*f*)/*I*^2^ is typically within a relatively narrow range of values 10^−6^–10^−7^ μm^2^/Hz at *f* = 1 Hz [60]. The observed results confirmed that back-gated graphene FET can detect a few gases using the FES method. It is possible because of the imperfections in the graphene structure, and it is responsible for the random adsorption–desorption process of ambient gas molecules. We can suppose that their decoration by noble metal nanoparticles can result in enhanced gas selectivity observed for MoS_2_ material [39] and, therefore, is worth more in-depth investigation.

The McWhorter model describes the low-frequency noise in MoS_2_ FETs well [65]. The charge traps, located at the MoS_2_ surface or the interface with the substrate, have distributed time constants and are responsible for the observed low-frequency noise. Its intensity decreases with the increasing number of MoS_2_ layers. The same effect is reached by annealing or passivation, mainly reducing the noise of contacts [66]. Noise changes in MoS_2_ FETs induced by ambient atmosphere were investigated for selected gases only. The changes of bias current, even by a few orders, were reported for a few gases but without observing Lorentzians, which are characteristic for gases chosen [30]. This material requires more detailed studies to determine the presence of Lorentzians related to the existing traps in the MoS_2_ structure at its doping or modulation by UV light.

### 2.3. Low-Frequency Noise Measurement Setups

Sensors are electronic devices that change their internal conduction mechanisms due to environmental changes. Consequently, their noise characteristics also change. Therefore, noise measurements, particularly low-frequency noise measurements, are a powerful tool to reveal such changes with higher sensitivity compared to other conventional approaches, which, by performing average measurements of electrical quantities, may not be able to detect individual microscopic physical or chemical events. Moreover, a gas sensor exposed to volatile agents can provide the same DC response. Still, we can reasonably expect that each chemical species interacts at a different rate and, possibly, with various active sites within the sensor matrix. These differences are likely to result in modified noise spectra shapes.

Two main aspects must be considered to optimize a measurement setup for conducting noise measurements on sensors: reducing the noise coming from the instrumentation and reducing the errors from the spectra estimation process [67]. Although standard instrumentation exists for low-frequency noise measurements (LFNM) laboratory setups and wafer-level device noise testing, dedicated instrumentation may be a better choice to maximize sensitivity and portability, depending on the sensor characteristics.

LFNM setups can be grouped into two main categories: voltage noise measurements (VNM) setups, usually employed for devices under test with a low or moderate impedance value, and current noise measurements (CNM) setups, to be preferred for high-impedance devices. Both kinds of setup are characterized by the same main building blocks, represented in Figure 4.

A bias network is fundamental; only thermal noise is present at its terminals if no bias is applied to the sensor. For the flicker noise to overcome the thermal noise, the sensor needs to be biased with a sufficiently large current. More precisely, since the power spectral density (PSD) of flicker noise is inversely proportional to the frequency, for any given bias level, there will always be a corner frequency below which the flicker noise is dominant and above which the flicker noise is negligible concerning the thermal noise. By increasing the bias, the corner frequency also increases, allowing observation of flicker spectra above a few hundred or a few thousand Hz, where accurate PSD estimation can be obtained in fractions of seconds [68]. However, it is not always possible to increase the bias at will. On the other hand, estimating the PSD at very low frequencies intrinsically requires much longer observation times. Therefore, there is a practical limit to the minimum observation frequency (typically from 0.1 Hz to 1 Hz, depending on the applications).

Additionally, the amplifier connected to the sensor introduces flicker noise. Therefore, the bias must be further increased so that the noise produced by the sensor overcomes the background noise (BN) of the measurement system. The bias network has to be designed not to introduce a significant level of inherent flicker noise that could be mistaken for the noise produced by the sensor. Using batteries is the most prominent and straightforward way to implement constant voltage supplies and—by connecting several batteries in series with a large resistor—low-noise stable current supplies. However, resorting to batteries limits the resolution to adjust voltages and currents. Continuous discharging also results in poor voltage stability over measurement time. Solid-state linear voltage regulators, even those marketed as “low-noise”, are generally too noisy, especially at low frequencies [69,70]. Very often, to overcome the limitations due to intense noise at the output of commercial programmable sources, dedicated systems need to be designed. In literature, a few low-noise current sources [71,72,73] (for VNM) and low-noise voltage sources [74,75] (for CNM) have been proposed.

The typical configuration of a low-noise current source with a very high equivalent output impedance employs the discrete low-noise junction field effect transistors (JFETs) connected as in Figure 5 [76]. The red blocks and connections in Figure 5 are absent in the most straightforward configuration, and the “voltage source” block is a battery. If a programmable current generator is needed, a few solutions are proposed. The source block is a floating voltage source employing solar cells [77]. The instrumentation amplifier (IA) and the control block (Figure 5), adequately designed, ensure good accuracy in adjusting the value of the supplied current with limited additive noise.

Concerning the possible designs of a low-noise programmable voltage source that overcomes the limitations of standard batteries, a solution based on a battery followed by a low-noise buffer as the reference source for an applied ladder network was proposed [78,79]. This approach offers very low noise and high initial accuracy, but it is affected by an output voltage drift due to the reference battery discharge. Moreover, the many components necessary for implementing a relay-based discrete resistors ladder network are expensive, complicated to mount, and challenging to copy. A second approach of a low-noise voltage source consists of filtering out the noise produced by a digital-analog converter (DAC) using a low-pass filter followed by a buffer. Black lines and blocks in Figure 6 represent this approach.

The filter (Figure 6) is an RC filter employing supercapacitors to attenuate inherent noise down to very low frequencies (f < 1 Hz). Supercapacitors have proven helpful for compact and low-noise filter design [80]. However, supercapacitors also suffer from a relatively large leakage current that could result in unacceptable errors in the value of the supplied voltage because of the voltage drop across the resistor needed to obtain the low-pass filter. Different solutions have been proposed to correct this error. These amendments are based on proper feedback configuration, where an error amplifier with an adjusted gain compensates for the difference introduced by the leakage current of the supercapacitor. One of these solutions is represented in Figure 6 and consists of adding the black scheme of the red block and connections (with the disconnected link between the DAC and the filter), as discussed in [75]. More recently, solutions with a microcontroller as a supervisor of the feedback loop have been proposed [81].

The first amplification stage is the most crucial in the setup design because it determines the background noise (BN) level that can be reached. In the case of voltage noise measurements at low input impedances, JFET input stages are preferred since those based on bipolar junction transistors (BJTs) do not offer good performances in the case of sensors with impedances above ~100 Ω due to higher values of their equivalent input current noise (EICN). When the EICN is negligible, it is the equivalent input voltage noise (EIVN) *e*_ivn_ (Figure 7) at the input of the amplifier that determines the BN and, therefore, the low-noise preamplifier is preferably designed by resorting to discrete large area Junction Field Effect Transistors (JFETs) for reaching low inherent noise levels. Indeed, suppose we resort to a commercial operational amplifier (OA) to design a low-noise voltage amplifier. In that case, the EIVN source can be coincident with the EIVN source of the OA. The JFET input operational amplifiers with the lowest level of equivalent input voltage noise (at low frequencies) that we have been able to find is the OPAx140 series by Texas Instruments [82], with an EIVN of about 50 nV/√Hz at 100 mHz, less than 16 nV/√Hz at 1 Hz, and about 5 nV/√Hz for the frequencies over 1 kHz. The discrete JFET of a large junction area, type IF3601 by InterFet [83], is characterized by an EIVN of 5.6 nV/√Hz at 100 mHz and 1.4 nV/√Hz at 1 Hz. When applying discrete JFET-based amplifiers, better noise performances require relatively more complex circuitry [84,85]. Still, an optimized design aimed at simplifying the implementation while maintaining excellent noise performances was demonstrated elsewhere [86].

For high impedances, current noise measurements are usually more easily implemented, and the preamplifier first stage is a low-noise transimpedance amplifier (LNTIA), as the ones reported in Figure 8. The OA used in these applications employ JFET or MOSFET-based input stages so that their EICN sources can be neglected [87]. In the most straightforward and conventional configuration of an LNTIA, the feedback impedance *Z*_f_ is a metallic film resistor with inherently low excess noise. Modified designs are possible in which the feedback impedance is a capacitor [88,89] that does not introduce noise. Also, approaches based on cross-correlation techniques that reduce the noise introduced by the feedback resistance [90,91] have been proposed. The ultimate level of BN at low frequencies is typically set by the thermal noise introduced by the feedback resistor. At the same time, the EIVN of the amplifier may become relevant either in the case of a noiseless feedback impedance or at higher frequencies when the impedance *Z*_sens_ of the examined sensor decreases because of capacitive components [90].

In the case of voltage noise measurements, an AC coupling filter for removing the DC component can be obtained with an R-C network with reasonable values (MΩ for the resistances and tens of μF at the maximum for the capacitances). The filter resistance must be much larger than the investigated *Z*_sens_ at all frequencies of interest to prevent attenuation of the measured sensor noise. Assuming that such resistance can be much larger than the sensor impedance, the coupling capacitance filters its thermal noise. A polyester or polypropylene high-voltage foil capacitor is typically applied. We must consider partial-discharge phenomena that generate interferences impacting the measured spectra and reduce the quality of noise measurements.

Noise contribution from the filter resistance must become negligible at the minimum frequency of interest *f*_min_. It means that the frequency corner of the AC filter must be much smaller than *f*_min_ [92,93]. The resulting long-time constant may cause long transients (in the order of minutes) if *f*_min_ is in the range of hundreds of mHz. It means that while exploring the lowest possible frequency can be interesting, the choice of the minimum frequency of interest must be made by considering the stabilization time one can afford in a given experimental setup.

In the case of low-resistance sensors (less than 10 Ω), signal transformers can be used in the measurement chain between the sensor and the preamplifier’s input to reduce the system’s background noise. A bridge arrangement is needed to bias the sensor with a constant current since coupling with a transformer requires zero residual DC. The bridge approach is still used almost exclusively in low-frequency noise measurements with transformer coupling. However, using a blocking capacitor would be, at least in principle, more straightforward and capable of lowering background noise. The most apparent reason why capacitors have not been used in this type of application is that even with magnetization inductances as large as tens of Hs, as it is not uncommon in signal transformers used in this type of measurement, achieving a resonance frequency well below 1 Hz would require blocking capacitors with capacitances above 0.1 F. Up to not many years ago, the capacitors of 1 F in a reasonable size and compatible with low-noise instrumentation were not available. Nowadays, supercapacitors that combine capacitances in the orders of a few Fs are available in small sizes and, more importantly, have been proven to be compatible with low-noise instrumentation [80]. The possibility of using supercapacitors instead of a bridge to solve the problem of a connection of biased devices to transformer-coupled low-noise amplifiers has been investigated [94]. The approach proposed in [94] provides better background noise performances and a more straightforward measurement procedure concerning conventional techniques based on a balanced bridge configuration. In such an approach, the bridge is eliminated. Therefore, the contribution to the noise coming from the balancing arm of the bridge is also eliminated. Moreover, without the bridge, the time-consuming balancing process is also excluded. The approach then results in faster measurements and higher sensitivity.

In the case of current noise measurement, an input AC coupling network capable of separating the DC bias current from the current fluctuations down to the mHz range would also be required. It could be conducted, in principle, by employing an L-R network that would operate dually concerning the R-C filter used for voltage noise measurement. In the case of current measurements, the resistance of the L-R filter must be much lower than the impedance of the investigated sensor for its attenuation effect to be negligible. This fact implies that the value of the inductance to obtain a low-frequency corner below 100 mHz would be more significant than 1 kH, which is outside the range of values that can be realized, at least with high quality and for signal applications. It means that the low-noise transimpedance amplifier is always coupled in DC.

### 2.4. Monitoring of Gas Exposure Conditions

Studies of gas sensor responses require a setup of calibrating gases and flowmeters to establish the necessary ambient atmosphere (Figure 9), comprising of a gas mixture diluted in synthetic air (S.A.—pure oxygen 20% mixed with pure nitrogen 80% without humidity). Resistance fluctuations can be measured when the gas flow is about a few hundred mL/min only to avoid gas turbulences influencing the results of the FES method. Calibrating gases can be mixed to establish the required gas composition using gas flow meters and homemade control software. The calibrating gases are delivered as a gas of interest diluted in nitrogen (N_2_). It can be further diluted in S.A., depending on the available gas flowmeters—the ratio of their gas flow ranges. We apply two sets of gas flowmeters to dilute the calibrating gas about three orders to reach tens of ppb concentrations for some gases. The simpler, cheaper, but much less accurate method of generating vapors uses a glass bubbler with a liquid compound evaporating because the flowing S.A. Gas concentration can be estimated depending on S.A. flow rate and the compound liquid’s vapor pressure [95]. The presented gas setup does not introduce humidity into the generated gas mixture, but a homemade humidifier can be applied to the gas setup to control humidity level precisely [96].

The calibrating gas setup is software-controlled to use the programmed changes of gas concentrations, synchronized with DC and flicker noise measurements (Figure 9A). The same design can control gas sensor modulation using their operating temperature changes or UV irradiation.

## 3. Results

### 3.1. Noise Power Spectral Densities of Selected Materials

The considered FES method was applied in chemoresistive gas sensors made of 2D materials. The detailed results were presented elsewhere [15,19,30,40,45,51]. The experimental studies used various noise measurement setups. We show the exemplary normalized current noise power spectral density *S*_I_(*f*)/*I*^2^ for two groups of materials having low DC resistance, not exceeding 20 kΩ, carbon materials (graphene, carbon nanotubes—Figure 10A), and semiconducting 2D materials (ZrS_3_, MoS_2_ ink-printed flakes) of DC resistances above a few hundred MΩ (Figure 10B). The first group of materials was investigated by applying a transimpedance amplifier (Figure 8), followed by an additional amplification stage. We observed low-frequency noise of various intensities in a measurement setup utilizing a battery supply and necessary shielding. Low DC resistance enabled the DC bias of the sensors with current I up to a few mA. The recorded noise was quite intensive, and the power supply interferences were visible in some measurements only. Semiconducting 2D materials were biased by a much lower current, typically tens or hundreds of nA at the maximum, which makes noise measurements more prone to power supply interferences. Fortunately, semiconducting 2D materials are noisy at low frequencies due to intense concentration fluctuations. We used low-noise amplification (Figure 7) of the noise signal *U*_i_ observed between the terminals of a low-noise metalized resistor (47 kΩ) in series with the semiconducting sensor biased by a stable voltage source. Voltage fluctuations across the resistor were proportional to current fluctuations induced by resistance noise in the semiconducting sensor. We could observe flicker noise at a low-frequency range, below 50 Hz, when the interferences of power supply lines were not visible (Figure 10B).

The normalized noise intensity *S*_I_(*f*)/*I*^2^ was much higher, up to a few orders, than for graphene or carbon nanotubes. All materials were irradiated to enhance their gas-sensing response. We observed the Lorentzian component for the graphene sensor at an ambient atmosphere of tetrahydrofuran (100 ppm) and a change in noise spectrum slope at a low-frequency range below 1 Hz for the MoS_2_ sensor. The change in flicker noise was observed in an ambient atmosphere of NO_2_ at 16 ppm. At lower NO_2_ gas concentrations, the changes were less stable due to some drift in time of the tested sensor. Better results should be observed when we resort to longer noise observation time and employ experimental setups that can allow us to separate drift in time from the change of flicker noise induced by ambient gas concentration. A recent thorough review of flicker noise measurements in various conditions can be found elsewhere [9]. Usually, flicker noise improves gas sensing at very low concentrations and saturates at higher values (e.g., H_2_S and NH_3_ gases detected by SnO_2_ gas sensor [12]), but the detailed behavior depends on the sensing material and ambient gas. We underline that the applied UV irradiation improves gas sensing and can replace thermal activation to some extent [97].

### 3.2. Classification Algorithms

Selected methods can analyze resistance noise data. Typically, we estimate the power spectral density of the recorded time series within the low-frequency range when 1/*f*-like noise prevails over the white noise component. The Welch method is used for power spectral density (PSD) estimation to reduce the random error of estimation using necessary spectra averaging [94]. The estimated power spectrum can determine Lorentzian frequencies or local slopes [7,10,98,99]. Next, the detection algorithm can determine the ambient atmosphere’s composition similarly to recorded DC resistance changes, using various chemometric or machine learning methods [100,101,102,103,104,105]. Before application, the algorithms require data reduction, eventually through a principal component analysis (PCA) method [106]. The PCA algorithm is well-known and commonly used for gas detection using DC resistances [107]. Although these methods are reliable enough, machine learning tools require intensive data processing, which implies a significant energy dissipation. Training machine learning and neural networks requires executing many measurements with various chemical compositions. Machine learning algorithms can provide better results than the SVM algorithm, and indeed, they have been successfully applied for noise power spectral data at selected conditions (modulated using UV irradiation [108]). These algorithms should be considered more often for analyzing noise spectra that are vectors comprising hundreds of correlated values. Machine learning gives a chance to utilize information about the ambient atmosphere included in noise power spectra more efficiently, especially for gas mixture detection.

Thus, much simpler and more direct approaches have also been tested with good results, for example, the binary [97,109] and ternary [11] fingerprint methods that extract a bit string from the measured PSD, generating a bit pattern characterizing the chemical agent that can be used as an address that directly calls the name of the chemical agent itself, requiring only a very low energy dissipation.

The summary of the considered detection algorithms is presented in Table 1. We consider the most popular algorithms successively applied within the last decades (e.g., PCA, SVM, and kNN) and recently developed machine learning methods (ANN) [101,108,110,111,112,113,114]. Additionally, we included the statistical methods, which can be used for noise time series parametrization (FES method) or as a detection algorithm based on the estimated spectra parameters of limited computational requirements [11,97,109].

The information delivered using the FES method about the ambient gas atmosphere is usually independent of DC resistances and, therefore, should be combined with the DC resistances to establish a data vector for detection algorithms. Moreover, the data for gas detection can be recorded at various gas sensors’ operating conditions (e.g., different elevated temperatures, UV irradiation) to enhance detection results, especially gas mixtures detected in practice [109,115].

The recorded noise time series can exhibit some drift due to unavoidable changes in gas sensors at their operation conditions, specifically by some pollutants (e.g., fungus and molds), pollens, and humidity in laboratory air. Therefore, some preprocessing to reduce eventual drift can be necessary [115,116,117]. Flicker noise time series can be used to determine their probability properties if different from a normal distribution (e.g., in the presence of a random telegraph noise component [118]). This remark is significant for 2D materials of small size when noise non-gaussian components can exist due to the tiny size of the sensing area and its 2D structure. We can consider various signal processing methods related to probability distribution (e.g., skewness, kurtosis, level-crossing statistics, etc.) [114,119,120,121]. Unfortunately, some statistical parameters of flicker noise require longer measurement time for averaging than power spectral density to reach the same random error of estimation.

## 4. Discussion

Practical use of the FES method in chemoresistive gas sensors requires adequate measurement setup for the selected gas sensors. Various flicker noise measurements are available. The systems focus on low-frequency measurements up to tens of kHz at the maximum for gas sensing applications. One of the essential issues is a reduction of the measurement system’s inherent noise. We aim to achieve it by applying the mentioned technical tips and commercial integrated OA circuits (e.g., OPA145, MAX4478) or discrete elements (e.g., JFET, type IF3601).

Additionally, we must consider the impact of external interference related to power lines (50 Hz and higher harmonics in Europe) or DC/DC converters emitting various frequencies related to their switching circuits, typically up to tens of kHz. The attenuation of low-frequency interferences can be reduced by the battery-powered setup, separating the measurement circuit from the externally conducted interferences and appropriate magnetic field shielding by using mumetal [122,123] or by applying lighter and easy-to-shape cobalt foil [124]. It is crucial to secure adequate grounding and avoid closed wire loops inducing interference sources in the measurement circuits. These remarks are essential for current noise measurements in gas sensors of high resistances when any wire works similarly to an antenna collecting external electromagnetic signals. We underline that some gas sensing materials are characterized by such high resistances (e.g., MoS_2_ [63,66]) that commonly used printed circuit boards can induce severe leakage currents. Teflon boards must be applied for their reduction.

Various noise mechanisms generate flicker noise observed in gas sensors of different materials (e.g., SnO_2_, WO_3_, CuO, graphene, MoS_2_, and ZrS_3_) [49,61,63]. We can expect different flicker noise intensity and its slope versus frequency. In the case of grainy gas sensing materials like SnO_2_, we observe 1/f noise generated by potential barrier modulation between the grains. Noise level is typically intense and can be relatively quickly recorded for the sensors of resistances not exceeding hundreds of kΩ. More problematic are the gas sensors made of a WO_3_ grainy layer, which can have a resistance of hundreds of MΩ at room temperature. Very high DC resistances also have the sensing layers of SnO_2_ at room temperatures, successively applied for NO_2_ detection [125]. Fortunately, elevated temperature and doping (e.g., by noble metal nanoparticles like Au and Pt) reduce its resistance and make noise measurements easier. It is worth mentioning that the applied UV irradiation induces a similar impact on gas sensing as thermal activation, including a decrease in DC resistance [97]. In the case of 2D materials or nanocrystals, UV irradiation is often more efficient than thermal activation because of more intensively increased photo adsorption at the surface [126,127]. Moreover, UV irradiation shortens the gas response times and lowers overall energy consumption.

Two-dimensional materials are highly interesting for gas sensing due to their extremely high active surface area-to-volume ratio. The FES method was effectively applied in these materials, but the studied materials exhibit different noise mechanisms. Mobility fluctuations generated flicker noise in the graphene layer, and its intensity was relatively small [63]. Noise in back-gated graphene FETs can be easily measured because such gas sensors are characterized by DC resistance of a few kΩ only. Additionally, low intensity of flicker noise induced by mobility fluctuation means that any gas molecule adsorption–desorption events can be observed as individual Lorentzians and used for different gas identification by a single gas sensor [15]. Two-dimensional materials with semiconducting properties are also promising for gas-sensing applications (e.g., MoS_2_ and ZrS_3_) due to the possibility of modulating their properties with bias conditions, operating temperature, or irradiation. These materials often have much higher DC resistance than graphene gas sensors. The observed flicker noise is generated mainly due to concentration fluctuations occurring in charge traps in their structural imperfections. It means that flicker noise is quite intense compared to graphene-based devices but more challenging to measure because of high DC resistances and low bias current. Moreover, when flicker noise is related to concentration fluctuations, it is more problematic to observe separate Lorentzians that enable the detection of different gases by a single gas sensor.

## 5. Conclusions

We discussed the methods of low-frequency noise measurements in resistive gas sensors. Chemoresitive sensors are popular, and the FES method applied in 2D materials is considered very promising for practical applications but requires a low-noise measurement setup. We divided the problem of low-noise measurements and spectra estimation into two cases: low and high sensor resistances. The designs were presented to reduce the inherent noise of the measurement setup in both cases. Generally, it is often easier to record the flicker noise of a gas sensor of lower resistance by observing voltage fluctuations. Gas sensors of high resistances require measurements of current noise prone to external interferences. We considered practical issues of noise measurements related to shielding, grounding, and noise spectra processing to determine the composition of an ambient gas mixture. Additionally, we underlined the impact of the noise mechanisms responsible for the observed flicker noise level. The presented systems and exemplary noise results further popularize the FES method in gas sensing by chemoresistive sensors.

## Figures and Tables

**Figure 1 sensors-24-00405-f001:**
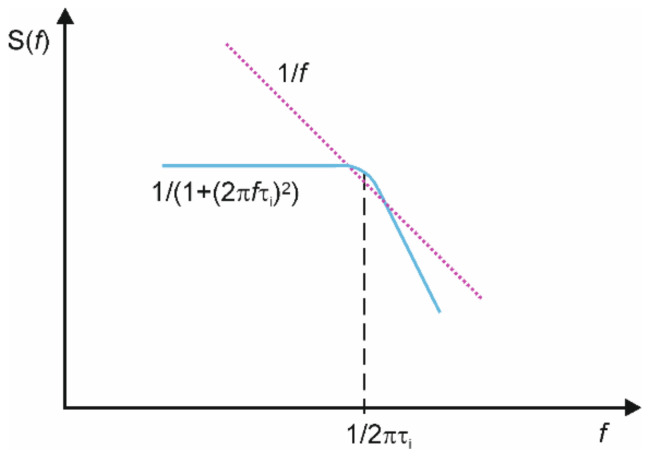
Schematic illustration of a plateau observed in 1/f-like noise power spectral density *S*(*f*) (pink dotted line) and generated by a single Lorentzian component *S*(*f*)~1/(1 + (2π*fτ*_i_)^2^ around the frequency 1/2π*τ*_i_ (blue solid line).

**Figure 2 sensors-24-00405-f002:**
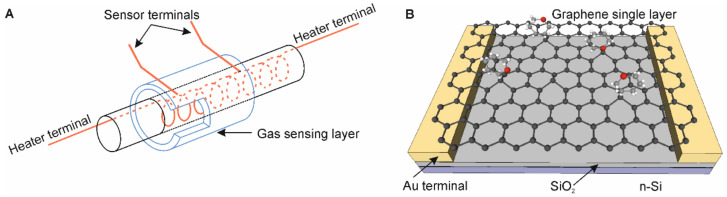
Illustration of resistive gas sensors: (**A**) comprised of grainy material operating at elevated temperature with a heater inside the gas sensing layer to reduce energy consumption, (**B**) graphene back-gated FET utilizing a single layer of graphene for better gas sensing with visible adsorbed ambient gas molecules.

**Figure 3 sensors-24-00405-f003:**
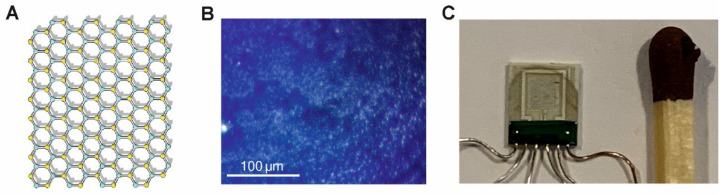
Exemplary gas sensor made of MoS_2_ flakes: (**A**) its chemical structure (S atoms—yellow color, Mo atoms—cyan color), (**B**) microscopic image of the ink-printed MoS_2_ aggregated flakes structure, (**C**) based on a ceramic platform (type KBI2, www.tesla-blatna.cz/de/senzory/, accessed on 1 January 2023) with golden interdigitated finger electrodes and heater with temperature sensor and prepared using ink-printed technology. The pristine flakes of MoS_2_ (www.graphene-supermarket.com, accessed on 1 January 2023; SKU: MOS2-100 ML), dispersed in ethanol-water solution, were sonificated to avoid aggregation for 30 min before multiple depositions on a platform by means of the precise fluid dispenser (Nordson (Westlake, OH, USA), type Ultimus Plus II), and next dried in laboratory air at the elevated temperature of 50 °C.

**Figure 4 sensors-24-00405-f004:**
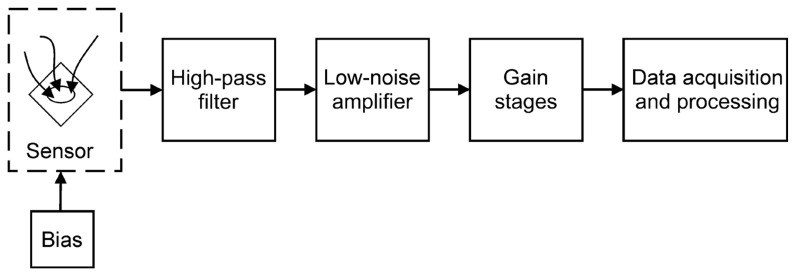
Schematic of a measurement setup for low-frequency noise measurements (LFNM).

**Figure 5 sensors-24-00405-f005:**
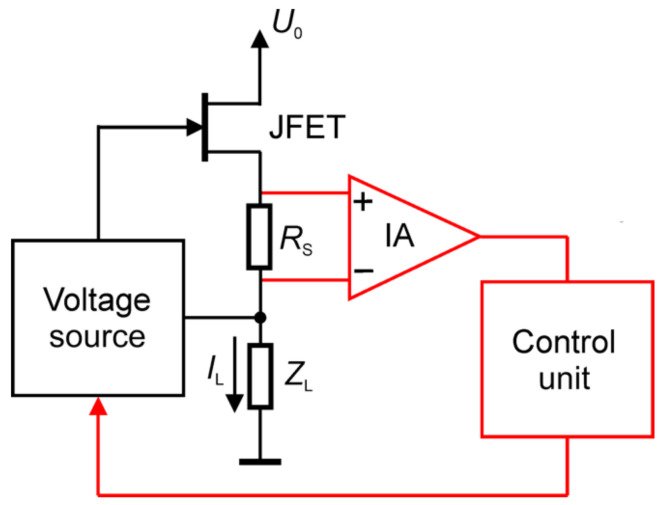
Configuration for low-noise current source utilizing discrete low-noise junction field effect transistor (JFET) biased by voltage *U*_0_ and the feedback to program voltage source by instrumentation amplifier (IA). The basic structure is black; the red blocks and connections are added to implement a programmable low-noise current source. The source delivers the current *I*_L_ for the examined sensor *R*_S_. The impedance *Z*_L_ is a loading impedance to the voltage source.

**Figure 6 sensors-24-00405-f006:**
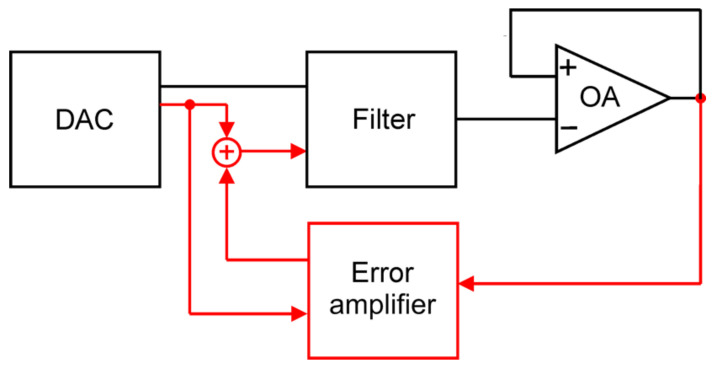
A low-noise voltage source is implemented by filtering out the noise produced by a digital-analog converter (DAC) employing a low-pass filter followed by an operational amplifier (OA) buffer (in black). More sophisticated solutions are reported in the literature, which, with appropriate feedback, reduce or eliminate any leakage errors due to the capacitance of the filter. An example is shown by adding the basic configuration with the red block and connections. The direct link between the DAC and the filter was removed in this case.

**Figure 7 sensors-24-00405-f007:**
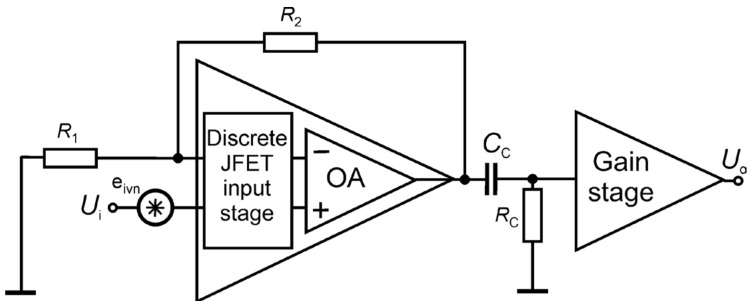
Schematic of the low-noise amplification stage.

**Figure 8 sensors-24-00405-f008:**
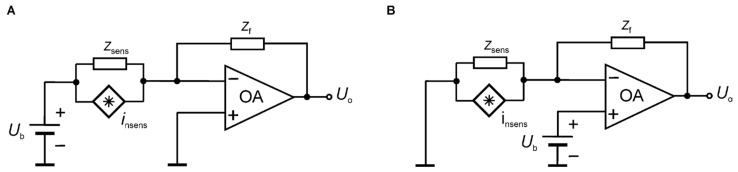
Schematic of a transimpedance amplifier with two different bias configurations: the solution in (**A**) supplies the direct current to the sensor; the solution in (**B**) supplies a small bias current at the input of the FET input OA.

**Figure 9 sensors-24-00405-f009:**
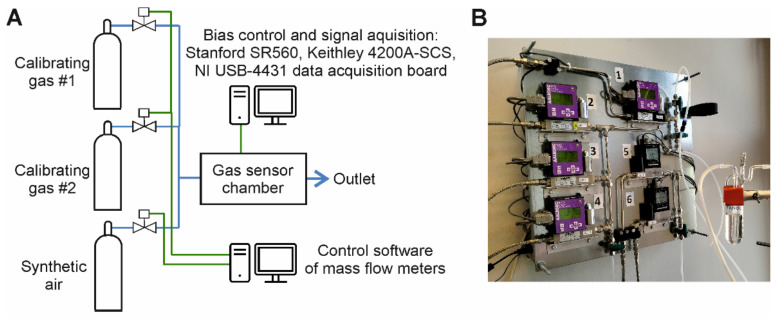
Exemplary setup of gas flow meters to generate gas mixtures by using the bottles of calibrating gases or a glass bubbler to create vapors of organic solvents (e.g., ethanol, methanol, chloroform, tetrahydrofuran, and acetonitrile): (**A**) schematic of gas installation, (**B**) a picture of the applied gas flow meters and a glass bubbler.

**Figure 10 sensors-24-00405-f010:**
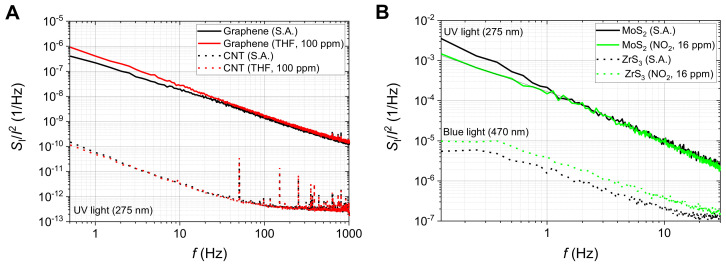
Exemplary power spectral densities measured for selected 2D materials: (**A**) back-gated graphene FET and carbon nanotubes (CNT) at an ambient atmosphere of synthetic air (S.A.) or tetrahydrofuran (THF) under UV irradiation (275 nm), (**B**) MoS_2_ ink-printed layer under UV irradiation (275 nm) and ZrS_3_ flake under blue light irradiation (470 nm) at an ambient atmosphere of synthetic air (S.A.) or NO_2_ diluted in S.A.

**Table 1 sensors-24-00405-t001:** Gas detection algorithms utilizing changes in DC resistances or noise power spectra.

Algorithm Name	Limitations and Strengths
Principal Component Analysis (PCA)	Popular, efficient for linear data, and less for long vector data [108].
Support Vector Machine (SVM)	Efficient for high dimensional data, including non-linear relation, requires more complicated computing and selection of kernel function with the adjusted parameters [106,110].
k-Nearest Neighbor (*k*-NN)	The method is gaining popularity as it combines ease of application with the ability to solve non-linear problems. Popular for mixed gases detection [112,113].
Artificial Neural Networks (ANN)	Accurate methods require adjusting the number of network layers. Easily computed using low-cost hardware. Numerous versions of ANN exist and can be applied [101,112]. These methods have the potential to become widespread.
Level crossing statistics	Operates on the recorded noise time series, easily realized with comparators and with low computation cost [114]. It can be applied for flicker noise with non-normal distribution.
Fingerprint methods in PSD analysis	Utilizes bandwidths and local slope of PSD [97,109,110]. It requires simple computing but can be used for the FES method only.

## Data Availability

The detailed data that support the findings of our experimental study are available from the corresponding author upon reasonable request.

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
