# Peer review of "Flicker Noise in Resistive Gas Sensors—Measurement Setups and Applications for Enhanced Gas Sensing"

_sensors, 2024, doi:10.3390/s24020405_

Round 1

Reviewer 1 Report

Comments and Suggestions for Authors

Major revisions:

The paper looks interesting. Below points may need further improving:

1. Fig.10: The work only studied the change of flicker noise with only gas concentration each time. Therefore, how about more gas concentrations? i.e could the change of flicker noise be correlated with different concentrations of gas? E.g. by measuring the flicker noise of MoS2 sensor to different concentrations of NO2 from 0.5 ppm tp 16 ppm with UV (275nm).

2. Fig.10: How about the thermal activation? Is it the same effect with UV? Need to study it.

3. Section 3.2 is not clearly presented. How was the classfication algorithms setup and the useful information presented? Show a specific example with data presented earlier in the paper .

Author Response

Dear Reviewer,

The detailed rebutal answer is included into the attached file.

With kind regards,

Janusz Smulko (corresponding author)

Reviewer 2 Report

Comments and Suggestions for Authors

In this manuscript (sensors-2806094), the authors studied low-frequency resistance noise measurements and their use for gas sensing by chemoresistive sensors. The research points are interesting, but there are some issues in writing, experiment, analysis, and discussion. The manuscript may be accepted after major modifications.

1. Introduction: (1) Suggest explaining the noise issue with specific examples (data). (2) Why take two-dimensional material gas sensors as an example? Metal oxide gas sensors operating at room temperature have even greater resistance such as Small, 2023, 19, 2303631.

2. Materials and Methods: Suggest simplifying the overview of resistive gas sensors and placing it appropriately in the introduction section.

3. Figure 3: The preparation method and material source of the gas sensor need to be explained.

4. Results: Lack of specific test results and data, such as response/recovery curves of the gas sensor at different gas concentrations.

5. The English writing and journal format need to be thoroughly revised. The numbers in the chemical formula require subscripts, including references. Figure and Fig., it needs to be unified as Figure.

6. Literature list: The authors cited many literature (over 100), but most of the literature is outdated. It is recommended to cite literature from the past three years, such as room temperature SnO2 oxide gas sensors.

Comments on the Quality of English Language

Minor editing of English language required.

Author Response

(The authors gave the same response as above.)

Reviewer 3 Report

Comments and Suggestions for Authors

The paper entitled “Flicker noise in resistive gas sensors – measurement setups and applications for enhanced gas sensing” describes the methods of low-frequency noise measurements in resistive gas sensors and the impact of noise mechanisms responsible for the observed flicker noise level.

The paper is precise and treat the noise issue in details. However, the authors do not bring a clear novelty. Here below are some points of improvement:

-        The article presents a review of the algorithms to treat the flicker noise. Despite the interesting work, the authors should add or combine algorithm concept to bring novel content in the literature. That would enhance the impact of the work.

-        The section “3.2. Classification algorithms” and/or “4. Discussion” would be clearer and easier for the reader with a comparative table of the algorithms with their limitation and strength.

Comments on the Quality of English Language

fine

Author Response

(The authors gave the same response as above.)

Round 2

Reviewer 1 Report

Comments and Suggestions for Authors

accept

Reviewer 2 Report

Comments and Suggestions for Authors

The response and revised manuscript are satisfactory, and it is recommended to accept.